# Natural World Distribution via Adaptive Confusion Energy Regularization

## Abstract

We introduce a novel and adaptive batch-wise regularization based on the proposed Batch Confusion Norm (BCN) to flexibly address the natural world distribution which usually involves fine-grained and long-tailed properties at the same time. The Fine-Grained Visual Classification (FGVC) problem is notably characterized by two intriguing properties, significant inter-class similarity and intra-class variations, which cause learning an effective FGVC classifier a challenging task. Existing techniques attempt to capture the discriminative parts by their modified attention mechanism. The long-tailed distribution of visual classification poses a great challenge for handling the class imbalance problem. Most of existing solutions usually focus on the class-balancing strategies, classifier normalization, or alleviating the negative gradient of tailed categories. Depart from the conventional approaches, we propose to tackle both problems simultaneously with the adaptive confusion concept. When inter-class similarity prevails in a batch, the BCN term can alleviate possible overfitting due to exploring image features of fine details. On the other hand, when inter-class similarity is not an issue, the class predictions from different samples would unavoidably yield a substantial BCN loss, and prompt the network learning to further reduce the cross-entropy loss. More importantly, extending the existing confusion energy-based framework to account for long-tailed scenario, BCN can learn to exert proper distribution of confusion strength over tailed and head categories to improve classification performance. While the resulting FGVC model by the BCN technique is effective, the performance can be consistently boosted by incorporating extra attention mechanism. In our experiments, we have obtained state-of-the-art results on several benchmark FGVC datasets, and also demonstrated that our approach is competitive on the popular natural world distribution dataset, iNaturalist2018.

## 1 Introduction

Fine-grained visual classification (FGVC) is an active and challenging problem in computer vision. Such a recognition task differs from the classical problem of large-scale visual classification (LSVC) by focusing on differentiating *similar* sub-categories of the same meta-category. In FGVC, the inter-class similarity among the object categories is often pervasive, while the intra-class variations further impose ambiguities in learning a unified and discriminative representation for each category. Long-tailed distribution brings in another aspect of challenge that the head categories tend to dominate the training procedure. The learned classification model thus performs better on these categories, while yielding significantly poor performance for the tail categories. The performance distribution somewhat resembles the data distribution. As the natural world distribution often assumes both fine-grained and long-tailed properties, how to satisfactorily address the recognition problem under such a general setting raises a practical and challenging problem.

From the existing literature, there are only a few attempts to solving these two problems at the same time. Relevant efforts mostly focus on tackling either task. In FGVC, most of the recent research efforts have converged to learn pivotal local/part details relevant to distinguishing fine-grained categories *e.g.*, (Fu et al., 2017; Yang et al., 2018; Zheng et al., 2019), and typically require the fusion of several sophisticated computer vision techniques to accomplish the task such as in (Ge et al., 2019). In resolving the long-tailed issue, previous approaches have looked into data balanced

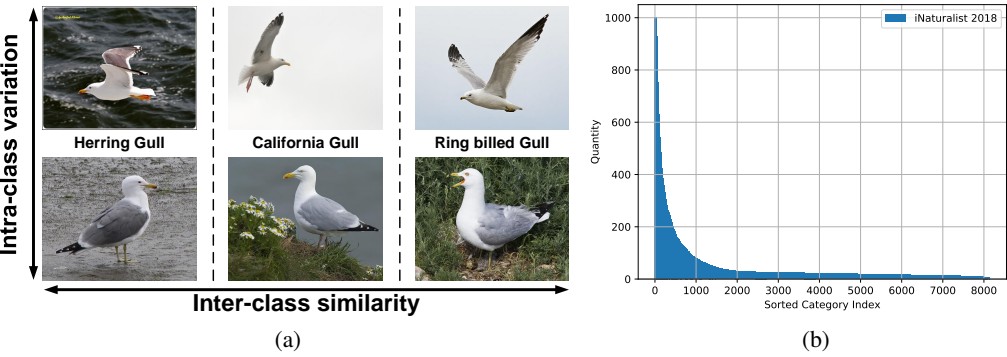

Figure 1: (a) Inter-class similarity vs. intra-class variation: Each column includes two instances of a specific "Gull" category from the CUB-200-2011 dataset Wah et al. (2011). (b) The natural world distribution dataset iNaturalist2018 Van Horn et al. (2018).

sampling (Huang et al., 2016a; Wang et al., 2017) and the recent development such as Kang et al. (2020) learns the representation at the first stage and refines the classifier by balanced data sampling.

Figure 1a illustrates the two aspects of paradoxes in FGVC where the inter-class similarity and the intra-class variations are subtly intertwined, yielding a daunting classification task. For humans, the example convincingly suggests that expert knowledge is needed to differentiate one from the other two categories. Alternatively, it also exhibits the challenges of formulating universal criteria in developing machine learning frameworks to satisfactorily solve the FGVC problem even for a modest case involving just three object categories. Figure 1b presents an extreme data distribution that some head categories have 1,000 images but only 2 images are included in a tailed category. Hence, a model by conventional training is expected to yield classification performance, displaying the long-tailed distribution on a balanced test/val set.

It goes without saying that techniques based on deep neural networks have been the focal point of the recent development in tackling FGVC. Characterized by powerful model capacity and end-to-end feature learning, these state-of-the-art approaches are craftily designed to extract discriminative local details and consistent global structure, and shown to achieve significant improvements over conventional non-DNN approaches, *e.g.*, Duan et al. (2012) on almost all FGVC benchmark datasets. However, the improvement for solving FGVC by exploring visual features of different levels and resolutions from relevant regions seems to be saturated and also does not properly address the long-tailed issue. The concern is reflected by that most FGVC methods do not include experimental results on the natural world distribution dataset iNaturalist2018 (Van Horn et al., 2018).

Motivated by these developments, we propose a flexible and effective regularization design that aims at guiding the resulting DNN learning to improve model efficiency on tackling the FGVC and long-tailed issues at the same time. Our method is relevant to the *pairwise confusion* regularization (Dubey et al., 2018); however, the proposed formulation goes beyond the restriction of working on pairs of data and develops a batch norm-based framework with sufficient model capacity to simultaneously deal with FGVC and long-tailed issues. We first assume all samples/images within a batch are of different classes. The targeted confusion energy is then modeled by a batch-wise matrix norm, termed as Batch Confusion Norm (BCN). The matrix is constructed by including prediction results from all images within a batch, as well as an adaptive matrix to adjust class-specific weights. The former is used to handle the FGVC task and the latter is for resolving the long-tailed distribution. To achieve efficient DNN learning, we provide an approximation scheme to BCN so that gradient backpropagation can be readily carried out. The promising experimental results support that BCN has good potential to function as a generic regularizer for solving a wide range of classification tasks.

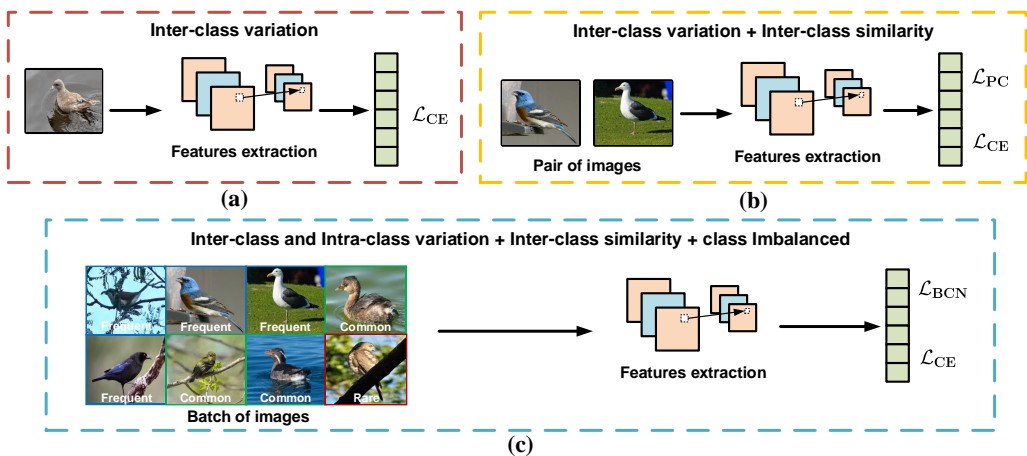

Figure 2: Learning to FGVC with (a) conventional Cross Entropy (CE) loss, (b) + Pairwise Confusion (PC) energy and (c) + the proposed Batch Confusion Norm (BCN).

## 2 RELATED WORK

Researches in fine-grained and long-tailed visual classification are going on in two different branches. Most articles focus on just one of these issues. We will introduce recent studies on both sides, and then briefly explain our approach.

**FGVC.** In the early works, the training data are annotated with additional information such as part labels. Along this line, Berg et al. (2014) explore the labeled part locations to eliminate highly similar object categories for improving the learned classifiers. The approach in Huang et al. (2016b) is established based on a two-stream classification network to explicitly capture both object-level and part-level information. However, owing to the rapid research advances in visual classification, the majority of recent FGVC approaches are designed to complete the model learning solely based on the information of category labels (Sun et al., 2019; Dubey et al., 2018; Wang et al., 2018; Li et al., 2018; Yang et al., 2018; Zheng et al., 2019; Chen et al., 2019).

**Long-Tail.** To alleviate the impact of the unbalanced data, the two common basic methods are re-sampling and re-weighting. In recently, a most common strategy is called class-balanced sampling (Shen et al., 2016). Different from instance-balanced sampling, every image has the same probability of being selected, class-balanced is to weight the sampling frequency of each image according to the number of samples of different categories. Furthermore, Gupta et al. (2019) proposed repeat factor sampling (RFS), a dynamic-sampling mechanism, to balance the instances. In general, re-sampling means that in the case of unbalanced existing data, the training samples that the model is artificially exposed to during learning are category balanced, so as to reduce the overfitting of the head data to a certain extent. Recently, Cao et al. (2019) introduces a label distribution aware margin loss that expands the decision boundaries of few-shot classes. Kang et al. (2020) decouples the learning procedure into two-stage, representations learning and classifier. And gives the conclusion that instance-balanced sampling gives more generalizable representations which can improve the performance after refining the classifiers by re-sampling.

**Confusion energy.** The confusion-related formulation for dealing with intra-class variations and inter-class similarity in FGVC have two main implications. First, it can be applied to alleviate the overfitting problem in training a FGVC model. Dubey *et al.*Dubey et al. (2018) construct a Siamese neural network, trained with a loss function including *pairwise confusion* (PC). The reasoning behind the design is that bringing the class probability distributions closer to each other could prevent the learned FGVC model from overfitting sample-specific artifacts. Second, the confusion tactic can be used to boost the FGVC performance by focusing on local evidence. Chen *et al.*Chen et al. (2019) partition each training image into several local regions and then shuffle them by a *region confusion*

*mechanism* (RCM). It implicitly excludes the information about the global object structure and forces the model to predict the category label based on local information. In other words, the ability of identifying the object category from local details is expected to be enhanced through shape confusion.

Our approach to FGVC and long-tail is most relevant to the above confusion-based approaches. We retain the advantages of confusion energy and exploit the potential in the long-tailed distribution. And then propose a novel confusion energy term called *Batch Confusion Norm* (BCN) which can flexibly adjust the confusion strength corresponding to the data distribution.

## 3 METHOD

The core of our method centers around the proposed BCN to not only alleviate the overfitting problem in training an FGVC model but also boost the classification performance. We begin by discussing the pairwise confusion energy (PC) (Dubey et al., 2018) and then elaborate the essential components of the proposed framework. We conclude the section with detailed explanations on how BCN is used to handle the long-tailed distribution dataset, namely, iNaturalist2018 (Van Horn et al., 2018).

### 3.1 BATCH CONFUSION NORM

Let $\Phi$ be the FGVC model as illustrated in Figure 2 and $\mathcal{D}$ be the training set over totally $C$ fine-grained categories. An arbitrary sample from $\mathcal{D}$ is denoted as $(\mathbf{x}, y)$ where in our case $\mathbf{x}$ is an image and $y \in \{1, \ldots, C\}$ is the corresponding class label. In learning $\Phi$, we follow the standard batch training and set the batch size to include $M$ images.

For each training sample $\mathbf{x}_i$ in a batch $\mathcal{B}$, the forward propagation through $\Phi$ would yield a class probability (*i.e.*, softmax) distribution $\mathbf{p}_i \in \mathbb{R}^C$. We can then define the batch-wise class prediction matrix by

$$P = [\mathbf{p}_1 \ \mathbf{p}_2 \ \cdots \ \mathbf{p}_M] \in \mathbb{R}^{C \times M}, \tag{1}$$

where each $\mathbf{p}_i$ is the softmax class prediction over the C fine-grained categories. Notice that the formulation assumes $M \leq C$ and all images within a batch $\mathcal{B}$ are randomly sampled. In contrast, the confusion regularization of PC Dubey et al. (2018) only affects the paired images with distinct labels.

An explicit purpose of BCN is to infuse *slight* classification confusions into the FGVC training procedure and drive the learning to work harder for making as many correct predictions in each training batch as possible. To this end, it is reasonable to minimize the rank of the batch prediction matrix $P$ so that all individual predictions are *similar*:

$$\underset{\Phi}{\arg\min} \operatorname{rank}(P). \tag{2}$$

However, the rank-related minimization problems are often NP-hard, and convex relaxations are instead introduced to approximate the solutions. In our formulation, minimizing the rank of $P$ is reduced to minimizing its *nuclear norm*. We define the *batch confusion norm* of $P$ as

$$\|P\|_{\text{BCN}} = \|P\|_* \tag{3}$$

where $\| \cdot \|_*$ is the nuclear norm that computes the sum of the singular values of the underlying tensor/matrix.

**Stability.** In order to make the matrix decomposition of $P$ being stable and to prevent the negative singular values affecting the training loss, we could replace the right-hand side of (3) with $\|P^\mathsf{T} P\|_*$ since it is known that

$$\operatorname{rank}(P) = \operatorname{rank}(P^\mathsf{T} P). \tag{4}$$

**Adaptability.** The BCN defined in (1) confuses each category evenly. However, the strategy does not take account of realistic data distribution such as long-tailed as well as fine-grained, and it could yield performance drop. Pertaining to those tailed classes of a few samples, infusing even small amount of confusion energy could easily degrade their classification outcomes. To resolve this issue, we include an adaptive matrix $A \in \mathbb{R}^{C \times C}$ into the model so that BCN can adjust the strength of confusion energy for each category. Here are a couple of criteria for initializing an appropriate $A$:

- When the data distribution is long-tailed, $A$ should alleviate the confusion energy on the tailed categories to prevent the model from getting excessive confusions over these classes.

- When the data distribution is balanced, $A$ should be almost the same as the identity matrix.

Follow these guidelines, we design the adaptive matrix $A$ as

$$A_{ij} = \begin{cases} (\log_{\mu+1}(\mathcal{N}_i + 1))^{\sigma^\tau}, & i = j \\ 0, & i \neq j \end{cases}, \tag{5}$$

where $\mathcal{N}_i, i \in \{1, 2, ..., C\}$, $\mu = \frac{1}{C}\sum_{i=1}^{C} \mathcal{N}_i$, $\sigma = \sqrt{\frac{1}{C}\sum_{i=1}^{C}(\mathcal{N}_i - \mu)^2}$, and $\tau$ are the size of each category, size mean, size standard deviation, and hyper-parameter, respectively. Note that when $\mathcal{N}_i \longrightarrow \mu$, we have $A_{ii} \longrightarrow 1$. Furthermore, if $\sigma \longrightarrow 0$ then $A_{ii} \longrightarrow 1$. This means that $A$ will reduce to the identity matrix when the data distribution is balanced.

Hence, to incorporate the batch confusion energy with the adaptive matrix $A$ into the total loss function for training, we have

$$\mathcal{L}_{\text{BCN}} = \|P^\mathsf{T} A^\mathsf{T} A P\|_{\text{BCN}}, \tag{6}$$

where the batch confusion loss $\mathcal{L}_{\text{BCN}}$ is computed based on the eigenvalues of $P^\mathsf{T} A^\mathsf{T} A P$.

**Learnability.** In practice, there is no feasible way to ensure that the parameters of $A$ given in (5) assume the optimal confusion. We instead use it as a *good* initialization and go for a learnable adaptive matrix, denoted as $\hat{A}$. Consequently, we modify the $\mathcal{L}_{\text{BCN}}$ into

$$\hat{\mathcal{L}}_{\text{BCN}} = \|P^\mathsf{T} \hat{A}^\mathsf{T} \hat{A} P\|_{\text{BCN}} + \eta\|\hat{A} - A\|_2, \tag{7}$$

where $\eta$ is the hyper-parameter to regulate that the learnable adaptive matrix $\hat{A}$ should not be too far away from $A$. Empirically, we initialize $\hat{A}$ with $A$ and set $\eta = 1$ to gain improvement over simply using the hand-crafted $A$ as the adaptive matrix.

## 3.2 Loss function

From (7) and the network architecture in Figure 2, the refined feature maps are followed by a fully-connected softmax layer to output the class prediction vector $\mathbf{p}$. The overall loss function can now be readily expressed by

$$\mathcal{L} = \mathcal{L}_{\text{CE}} + \lambda\hat{\mathcal{L}}_{\text{BCN}} \tag{8}$$

where $\mathcal{L}_{\text{CE}}$ is the cross-entropy loss which is usually applied in classification task and $\lambda$ is a hyper-parameter to adjust the influence of the BCN loss to learning the model.

## 4 Experimental results

We conduct extensive experiments to evaluate our approach on three balanced benchmark FGVC datasets and the natural world distribution dataset. We then describe comparisons to prior work as well as the implementation details. We also provide an insightful ablation study for assessing the performance gains of using adaptive confusion energy BCN. Finally, a number of visualization examples are demonstrated for further discussions.

## 4.1 Datasets

We first evaluate the effectiveness of the proposed approach on three standard fine-grained visual classification datasets, namely, CUB-200-2011 (Wah et al., 2011), Stanford Cars (Krause et al., 2013), and FGVC-Aircraft (Maji et al., 2013). Table 1 shows the detailed statistics with the numbers of training and testing splits along with the category numbers. The data ratio between the training and the testing is about $1:1$ for CUB-200-2011 and Stanford Cars, and is about $2:1$ in FGVC-Aircraft. The class distribution of the three datasets is nearly balanced which can be used to measure the performance of the proposed method only in the fine-grained scenario with adaptive matrix $\hat{A}$

Table 1: Statistics of 3 balanced FGVC datasets and the natural world dataset iNaturalist2018.

| Dataset | CUB | CAR | AIR | iNaturlist2018 |
|---|---|---|---|---|
| # Train | 5,994 | 8,144 | 6,667 | 437513 |
| # Val/Test | 5,794 | 8,041 | 3,333 | 24426 |
| # Category | 200 | 196 | 100 | 8142 |

approximating identity matrix. Compared with other datasets for the large-scale visual classification task, these three FGVC datasets have obviously fewer training data for each category.

We then focus on the natural world distribution dataset iNaturalist2018 (Van Horn et al., 2018) which has the properties of both fine-grained and long-tailed distribution. Besides, it is also a large-scale dataset. Judging from the recent literature (Cao et al., 2019; Kang et al., 2020), this is a fairly challenging dataset that the performance can serve as an objective measure about the usefulness of our method. Finally, we remark that the proposed model does not require any additional annotations in the training process but merely the image-level class annotations.

## 4.2 IMPLEMENTATION DETAILS

We describe the implementation details with FGVC and iNaturalist2018. All our inference results are obtained from end-to-end training. We implement our method by using Pytorch framewrok (Paszke et al., 2017), and the source code will be made available online.

**FGVC.** Following relevant work (Yang et al., 2018; Chen et al., 2019; Zheng et al., 2019), we evaluate our method on the widely-used classification backbone ResNet-50 (He et al., 2016) which is pre-trained on the ImageNet dataset. For the sake of fair comparison in FGVC training, we use the data augmentation setting as in Chen et al. (2019) that the input size is set as $448 \times 448$, and horizontal flipping is randomly performed. The initial learning rate and the hyper-parameter $\lambda$ are 0.008 and 10, respectively. The training batch size usually is 16 if the GPU memory is enough and the training optimizer is Momentum SGD, which accompanies with cosine annealing (Loshchilov & Hutter, 2017) as the learning rate decay.

**iNaturalist2018.** We further evaluate the proposed BCN on the iNaturalist2018 dataset. In addition to using similar augmentation schemes, we set up the training conditions as in Kang et al. (2020); Cao et al. (2019) that the backbones, input size, training epochs, optimization method, and learning rate schedule are ResNet-50/ResNet152, 224, 90, SGD, and cosine annealing, respectively.

**Evaluation.** After training on the FGVC and natural world datasets, we evaluate the models on the corresponding balanced test/validation datasets and report the top-1 accuracy which is used commonly. The value of accuracy is reported in the format of percentage.

## 4.3 PERFORMANCE OVERVIEW

The proposed BCN not only preserves the benefits of confusion energy in FGVC task, but also addresses the downside of the confusion energy in long-tailed challenge. Table 2 shows the overall results on three benchmark FGVC datasets and the natural world distribution dataset. In FGVC, we have developed a general way to explore confusion energy (compared with PC). In Table 2a, we compare the results with recent state-of-the-art approaches on ResNet-50 backbone (Sun et al., 2019; Dubey et al., 2018; Wang et al., 2018; Li et al., 2018; Yang et al., 2018; Zheng et al., 2019; Chen et al., 2019). We further boost the performance with a simple attention mechanism GASPP to make BCN more competitive to the state-of-the-art approaches. The details of GASPP can be found in Appendix B.1. Table 2b includes the results on iNaturalist2018. We see that when adaptive matrix is set to the identity matrix $I$, the use of confusion energy drops the performance in both PC and BCN. The performance drops indicate that the adaptive matrix $\hat{A}$ plays a pivotal role in solving the long-tailed problem. Indeed, $\hat{A}$ enables BCN to just focus on the head categories but alleviate the

Table 2: Accuracy (%) on the test sets of three FGVC datasets and the validation set of iNaturalist2018. The superscript † means that the models are re-implemented by the same training setting to ours.

<table>
<tr><td colspan="4" align="center">(a)</td><td colspan="3" align="center">(b)</td></tr>
<tr><td>Method</td><td>CUB</td><td>CAR</td><td>AIR</td><td>Method</td><td>ResNet-50</td><td>ResNet152</td></tr>
<tr><td>PC†</td><td>87.0</td><td>93.5</td><td>92.4</td><td>CB-Focal</td><td>61.1</td><td>-</td></tr>
<tr><td>DB</td><td>87.7</td><td>94.3</td><td>92.1</td><td>LADM (SGD)</td><td>64.6</td><td>-</td></tr>
<tr><td>DFL-CNN</td><td>87.4</td><td>93.1</td><td>91.7</td><td>Baseline</td><td>61.7</td><td>65.0</td></tr>
<tr><td>NTS-Net</td><td>87.5</td><td>93.9</td><td>91.4</td><td>cRT</td><td>65.2</td><td>68.5</td></tr>
<tr><td>DCL</td><td>87.8</td><td>94.5</td><td>93.0</td><td>LWS</td><td>65.9</td><td>69.1</td></tr>
<tr><td>TASN</td><td>87.9</td><td>93.8</td><td>-</td><td>PC†</td><td>61.4</td><td>64.1</td></tr>
<tr><td>iSQRT-COV</td><td>88.1</td><td>92.8</td><td>90.0</td><td>Ours ($\hat{A} = I$)</td><td>61.6</td><td>64.7</td></tr>
<tr><td>Ours</td><td>87.8</td><td>94.3</td><td>93.2</td><td>Ours</td><td>**66.1**</td><td>**69.5**</td></tr>
<tr><td>Ours + GASPP</td><td>**88.4**</td><td>**94.7**</td><td>**93.5**</td><td></td><td></td><td></td></tr>
</table>

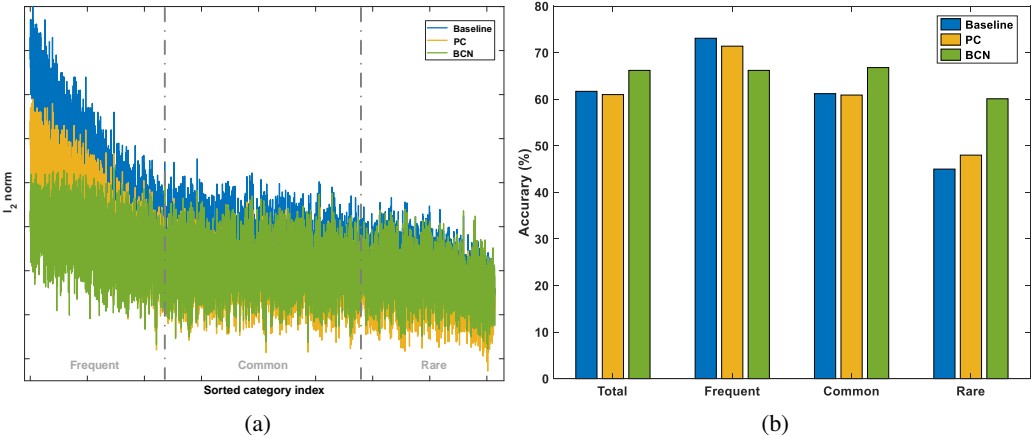

(a)  (b)

Figure 3: Baseline versus confusion energy models. (a) The $l_2$-norm of each category corresponds to the weight $\mathbf{w}_i$ in the classifier. (b) The classification accuracy of *Frequent*, *Common*, and *Rare* from the baseline and the other two confusion energy methods, PC and BCN.

confusion energy effect on the tailed categories. Note that, our models are trained not only with the most common way of data sampling *instance-balanced sampling* but also end-to-end. In contrast, Kang et al. (2020) trains the model in two stages and requires the use of *class-balanced sampling*.

## 4.4 ANALYSIS

Dubey et al. (2018) has shown that confusion energy alleviates the overfitting problem and improves the FGVC performance. However, we observe that if the baseline model coupled with the confusion energy directly, the overall performance will drop on the natural world dataset. It suggests that the long-tailed issue needs further investigations beyond the conventional model of confusion energy. Consider next the magnitude of each category corresponding to the classifier weight $\mathbf{w}_i$ in Figure 3a. The scale of $\|\|\mathbf{w}_i\|\|$ distribution on the baseline method is very similar to the data distribution. In addition, although PC has alleviated the scale of the head categories, but the distribution does not change significantly. Note that BCN ($\hat{A} = I$) has the similar phenomenon as PC. Nevertheless, the adaptive confusion energy BCN, makes the scale of the head to become smoother. This means that the prediction of the classification will not be dominated by the weights of head categories. We

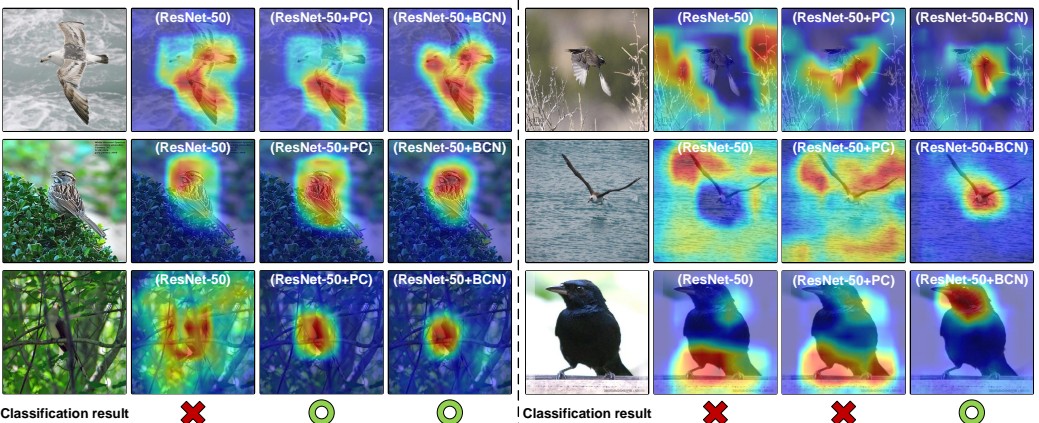

Figure 4: Heatmap-visualization of testing images by Grad-CAM (Selvaraju et al., 2017). The spatial heatmaps show the responses of the network to different images. For each image set, the first column shows the input images; the remained three columns show the corresponding heatmap of each model.

further split the categories into three groups, *Frequent* ($\mathcal{N}_i >= 100$), *Common* ($\mathcal{N}_i >= 10$), and *Rare* ($\mathcal{N}_i < 10$), and evaluate the performance of each group. The results are presented in Figure 3b, where the conventional confusion energy still behaves like the baseline, but BCN yields a more uniformly distributed performance.

## 4.5 DISCUSSIONS

In summary, BCN provides several benefits. First, it alleviates the overfitting problem of the cross-entropy loss. While training with the cross-entropy loss concerning the ground truth label in the manner of the one-hot vector, the inter-class similarity information is usually significantly suppressed. Consequently, it is not able to capture the fine-grained essence by one single cross-entropy loss while handling the overfitting issue. The proposed BCN successfully alleviates this issue. Second, BCN forces the model to learn the inter-class similarity so that the classifier is more focused on the discriminative parts. This phenomenon can be found by using the class activation mapping (Grad-CAM) (Selvaraju et al., 2017) presented in Figure 4. Without any attention mechanism and additional annotation, we successfully make the attention region smaller and accurately attend on the discriminative parts only by BCN. Third, BCN does not require additional processing of inputs and outputs during training and there is no extra cost at inference time, which makes it flexible and applicable to real applications. Final, BCN solves the confusion energy problem while meets the long-tailed distribution. BCN coupled with the adaptive matrix $\hat{A}$ not only preserves the benefits of confusion energy in FGVC task, but also addresses its downside in long-tailed scenario.

## 5 CONCLUSIONS

We have developed a general regularization technique specifically designed for addressing the fine-grained visual classification and the long-tailed data distribution problems simultaneously. The proposed Batch Confusion Norm (BCN), together with the standard cross entropy loss can be used to account for the inherent classification difficulties due to inter-class similarity and intra-class variations. And also solves the long-tailed problem by an adaptive matrix term. The proposed BCN considers the confusion regularization within each training batch and thus is more general than the relevant formulation of pairwise confusion energy. The resulting model is shown to be capable of learning discriminative features within regions of interest and alleviating the overfitting problem in training. The provided experimental results nearly achieve state-of-the-art over the three mainstream FGVC datasets and are competitive to leading long-tailed approaches on the natural world distribution dataset. Our future work will focus on generalizing the adaptive BCN concept to tensors and also on extending its applications to other challenging computer vision problems.

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

APPENDIX

In this section, we provide some supplementary information to explore how confusion energy regularization term works in FGVC task. And then present some interesting phenomena that the conventional methods does not have.

## A    ABLATION STUDY

Table 3: FGVC accuracy comparisons on the standard FGVC datasets CUB-200-2011 (CUB), Stanford Cars (Cars), and FGVC-Aircraft (Aircraft).

| Model | ResNet-50 | | | ResNeXt-50 | | | ResNeXt-101 | | |
|---|---|---|---|---|---|---|---|---|---|
| | CUB | CAR | AIR | CUB | CAR | AIR | CUB | CAR | AIR |
| baseline | 85.5 | 92.7 | 90.3 | 86.3 | 93.1 | 90.9 | 87.3 | 93.5 | 91.6 |
| baseline + PC | 87.0 | 92.4 | 90.1 | 87.5 | 93.2 | 91.2 | 88.2 | 93.7 | 92.4 |
| baseline + BCN | 87.8 | 94.3 | 93.2 | 88.1 | 94.4 | 93.3 | 88.6 | 94.5 | 93.5 |

To investigate the performance of different confusion energies between the different backbones, we make an ablation study on the ResNet-50, ResNeXt-50, and ResNeXt101. Table 3 shows the ablation between PC and BCN. The confusion energies both obviously improve the baseline performance on the CUB. Since CUB is the most difficult in the three benchmark datasets. But while the datasets are more easier, although BCN gains more improvement, the confusion energies provide little help. And take a look at the FGVC researches recently, it seems to have reached the limitation so far. Hence, it is reasonable to go through the more challenge task, fine-grained and long-tailed.

## B    ARCHITECTURE DETAILS

Let $F \in \mathbb{R}^{c \times h \times w}$ be the feature maps obtained by the last convolutional layer of the ResNet backbone as shown in Figure 5. We extend the network with two streams: one for learning discriminative features and the other for uncovering the proper attention responses. In both streams, we use the *atrous spatial pyramid pooling* (ASPP) technique to simultaneously extract features/attentions from different field-of-views.

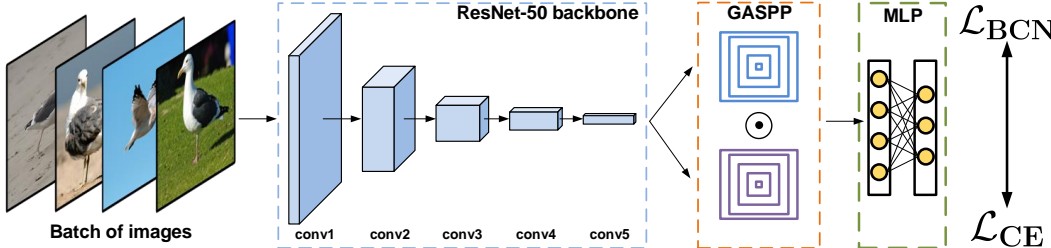

Figure 5: The proposed neural network architecture $\Phi$ for tackling the task.

### B.1    ATTENTION GATED ASPP

In addition, our method is implemented with an attention-gated network, boosted with the use of Atrous Spatial Pyramid Pooling (ASPP) technique. As shown in Figure 6, in the feature stream, performing ASPP would covert $F$ into $\mathrm{ASPP}_f(F)$, while in the attention stream, the similar procedure would yield attention feature maps, $\mathrm{ASPP}_a(F)$. Note that parameters in the two streams are not shared but learned jointly. After regulating with respect to the respective activation function, we carry

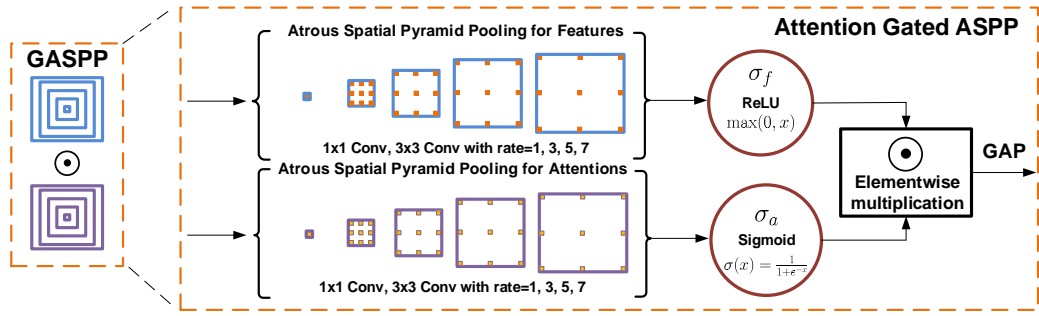

Figure 6: The attention gated ASPP architecture

out the gated element-wise product to output the adjusted feature maps $\tilde{F}$, weighted by the predicted attentions. That is, the attention-gated ASPP feature maps $\tilde{F}$ are derived by

$$\tilde{F} = \sigma_f(\mathrm{ASPP}_f(F)) \odot \sigma_a(\mathrm{ASPP}_a(F)) \tag{9}$$

where $\tilde{F} \in \mathbb{R}^{c \times h \times w}$ remains the same dimensions, $\odot$ denotes the element-wise product, $\sigma_f$ in our implementation is ReLU and $\sigma_a$ is the sigmoid function for gating. The ASPP operation is similar to that in Chen et al. (2017) except that we use different dilated rates. In summary, $\mathrm{ASPP}_f(\cdot)$ would learn the image features across the whole spatial domain and $\mathrm{ASPP}_a(\cdot)$ instead predicts the attention heatmaps. The gated fusion between the two streams leads to the output of discriminative feature maps $\tilde{F}$ for FGVC.

## C  BCN REASONING.

To justify the design of BCN, we use Figure 7 to illustrate the underlying mechanism. As we have mentioned before, the complexity of FGVC originates from the dilemma of simultaneously addressing the inter-class similarity and intra-class variations. Our formulation requires the samples in each training batch to be of different labels, and thus leads to two extreme cases to be considered. When the inter-class similarity in a batch is significant, as shown in Figure 7a, the BCN loss can be considered as a typical regularization term to avoid overfitting. On the other hand, when inter-class similarity is not of concern as in Figure 7b, the distributions of class prediction from the samples in a batch could vary significantly and induce a substantial BCN loss. Thus, the training would explore discriminative features to reduce the cross-entropy loss and consequently boost the FGVC performance.

## D  THE ADVANTAGE OF BCN

To investigate the advantage of the proposed BCN, we show the prediction layer with *softmax* activation at Figure 8. In standard deep neural networks for classification, the loss function for prediction layer is cross-entropy. Other methods such as NTS-Net Yang et al. (2018), DCL Chen et al. (2019) and TASN Zheng et al. (2019) adopt many loss functions, but their loss function for the prediction layer remains the same. If only the cross-entropy is used to learn the prediction layer, the output probability tends to be very close to one-hot vector even though the prediction is not correct. We highlight the observation in Figure 8. This outcome conflicts the real scenarios in FGVC problems. In FGVC, the categories sharing high similarities should have similar classification probabilities to recognize them but not concentrating on a specific category as one-hot vector. With the help of BCN, which already learned the characteristics of FGVC, the satisfactory results are expected.

Figure 9b shows the influence of confusion regularization in respect of the training loss. This experiment shows that training our model in various value of $\lambda$ has similar convergence speeds but has noticeable saturated training losses. The higher weight of confusion regularization brings the larger training loss, and the larger training loss means the model is harder to over-fit the training data. Notice that the standard FGVC datasets often have a small number of training images for each category; hence, it is easy for a deep neural network to over-fit the training data. Figure 9a presents that with the confusion regularization term, the entropy of prediction $\mathbf{p}_i$ would be larger than the one

## Dual-effect of BCN loss: $\mathcal{L} = \mathcal{L}_{\mathrm{CE}} + \lambda\mathcal{L}_{\mathrm{BCN}}$

### (a) Heavy inter-class similarity Batch: BCN alleviates overfitting

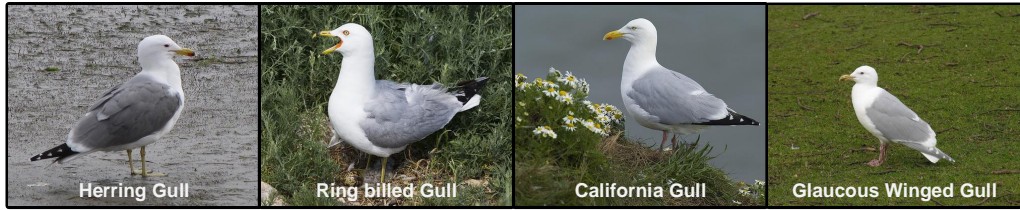

### (b) Moderate inter-class similarity Batch: BCN improves FGVC in tail classes

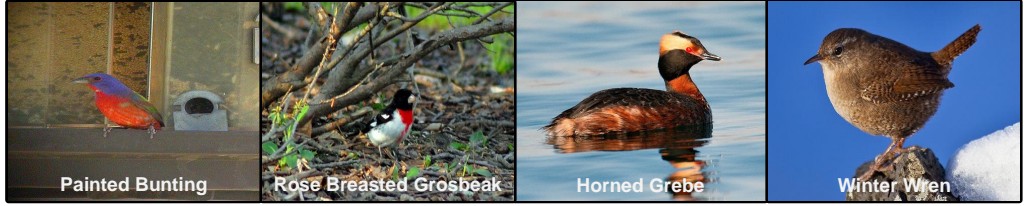

Figure 7: All images in a training batch are of different class labels. **(a)**: When inter-class similarity prevails in a batch, the BCN loss functions as regularization to avoid overfitting. **(b)**: When inter-class similarity is not obvious, the batch would yield a significant BCN loss and the optimization would turn to further reduce the cross-entropy loss and thus is expect to boost the FGVC performance in inference.

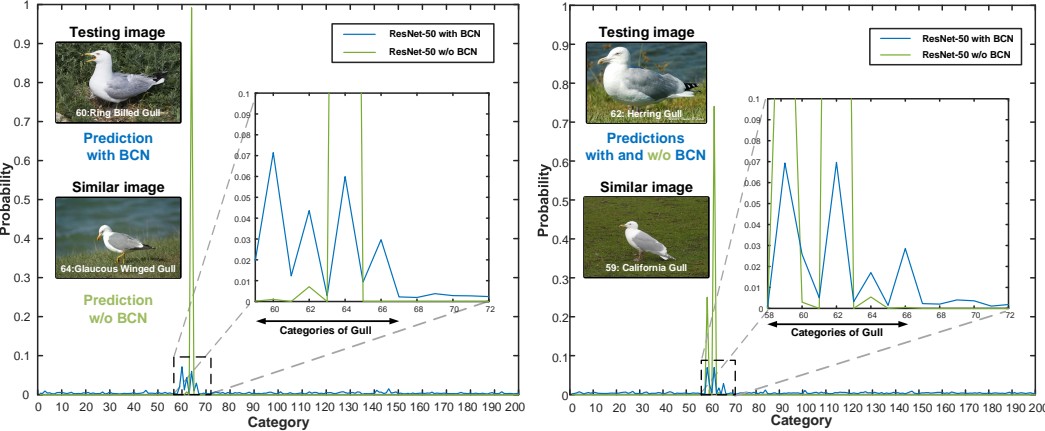

Figure 8: Prediction-visualization of FGVC. The X-axis denotes the category index, and the Y-axis denotes the classification confidence. This figure shows the predictions of ResNet-50 with/without coupling BCN. Blue lines indicate coupling the proposed BCN with the vanilla ResNet-50, which correctly classifies the two testing images. Green lines indicate the predictions of the vanilla ResNet-50, which only correctly classify the right testing image. One similar image is attached under each test image. With the aids of BCN, the classification model try to exert a regularization effect from other images in the same batch, hence shows hesitation among similar categories. However, the model trained without confusion loss seems very confident even though making a wrong prediction.

without in inference. The model has learnt the inter-class similarity and needs to make the decision from the discriminative points. For The black line in Figure 9b shows that it is easy to over-fit the CUB-200-2011, Standord Cars and FGVC-Aircraft datasets using ResNet-50 without applying BCN or PC. Therefore, the proposed BCN and previous PC can be used to reduce the over-fitting issue in the FGVC task.

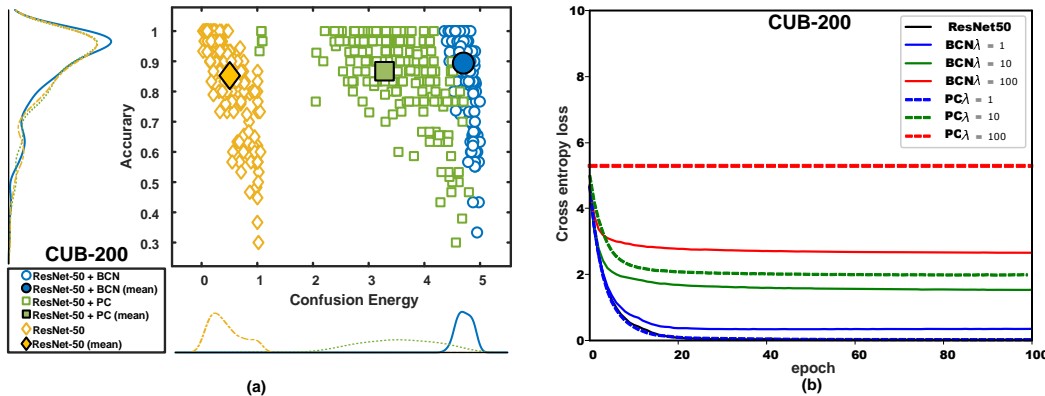

**(a)** **(b)**

Figure 9: The impacts by BCN and PC on CUB-200-2011. **(a)**: The regularization terms promote the performance in the FGVC task by confusion energy. Each point represents a class in the task. The BCN is more stable than PC, giving each class similar confusion energy, which eventually leads to more classes with higher accuracy. **(b)**: The training CE loss with different $\lambda$ values. The hyper-parameter $\lambda$ is used to control the weight of BCN and PC in the loss function. According to these figures, we observe that the phenomenon is similar in each data set.

Take attention to the situation with $\lambda = 100$. In Figure 9b, while BCN provides valid predictions, the PC destroys the entire training process. This indicates that BCN is insensitive and stable to the influence of hyper-parameter $\lambda$. At the same time, there is a wide range between 10 and 100 but the performance is very similar. So we don't have to perform manual tuning. In addition, when the hyper-parameter $\lambda$ is greater than 1.0, the PC may provide better results, but it may also cause the model to be completely untrainable.

