# OpenReview forum: "Natural World Distribution via Adaptive Confusion Energy Regularization"
_ICLR.cc/2021/Conference — Reject_

### Official Review · AnonReviewer3 · 2020-10-27
**Interesting idea, poor presentation**

**Rating:** 4
**Confidence:** 3

**Review:**

**Summary:**
This paper is about fine-grained visual classification, which is challenging due to high inter-class similarity, high intra-class variation and potentially also class imbalance. The authors propose a regularization term that is added to a standard cross-entropy loss of a neural network trained in a supervised fashion. This regularization shares the motivation with prior art to confuse the network and reduce over-fitting by making all predictions within a mini-batch similar to each other. Different to prior art, they authors propose an approximation of a minimum rank objective. To also handle class imbalance, the authors extend the regularization with a learnable matrix that can automatically balance the importance of individual classes.

**Pros:**
- Overall, I think the contributions of the paper is interesting and useful, particularly the extension of the confusion-based regularization for class-imbalanced data sets. Unfortunately, the presentation of the idea needs significant improvement, see below.
- The analysis and discussion in Sections 4.4 and 4.5 are great

**Cons:**
- The writing made this paper really hard to understand. The formulations, particularly in abstract and introduction, are inaccurate and vague. They should rather be specific and concrete. For instance, the statement "When inter-class similarity prevails in a batch, the BCN term can alleviate possible overfitting due to exploring image features of fine details" is hard to understand since it was totally unclear what the BCN term actually is at that point of reading. In both abstract and introduction, I had a very hard time imagining what certain statements mean in terms of who the method would look like, without having read the full paper.
- The results barely improve over SOTA, particularly for the three FGVC data sets. So the bigger advantage I see was for class imbalanced data sets, like iNaturalist. Although the performance is also not significantly better than prior works, the direct competitor (PC) seems to under-perform clearly. The proposed solution alleviates the problem, which is good. In light of this, I think it would make sense to build such data sets synthetically from the existing ones (like Cub, Car, Air) by removing samples to increase class imbalance. This would allow additional experiments on class imbalance in a controlled setup.


**Minor notes:**
- The author names can be dropped with the ICLR citation format: "... Dubey et al.Dubey et al. (2018) construct a Siamese" on page 3.
- Statements like "The confusion-related formulation for dealing with intra-class variations and inter-class similarity in FGVC have two main implications" on page 3 require the reader to be very familiar with these concepts. It would be better to re-formulate it with a brief introduction of the concept (just a sentence or two).

---

> ### Author Response · Authors · 2020-11-25
> **We will revise some sentences to make it easier to follow.**
>
> [Q1]. The writing made this paper really hard to understand. The formulations, particularly in abstract and introduction, are inaccurate and vague. They should rather be specific and concrete. For instance, the statement "When inter-class similarity prevails in a batch, the BCN term can alleviate possible overfitting due to exploring image features of fine details" is hard to understand since it was totally unclear what the BCN term actually is at that point of reading. In both abstract and introduction, I had a very hard time imagining what certain statements mean in terms of who the method would look like, without having read the full paper.
>
> [A1]. We are sorry for the problem. But here is the reason. We straddle two different domains at the same time, and each of them is the mainstream research direction recently. Hence, due to the limited space, we should try to write the article concisely. We will review and revise the structure we write.
>
> [Q2]. The results barely improve over SOTA, particularly for the three FGVC data sets. So the bigger advantage I see was for class imbalanced data sets, like iNaturalist. Although the performance is also not significantly better than prior works, the direct competitor (PC) seems to under-perform clearly. The proposed solution alleviates the problem, which is good. In light of this, I think it would make sense to build such data sets synthetically from the existing ones (like Cub, Car, Air) by removing samples to increase class imbalance. This would allow additional experiments on class imbalance in a controlled setup.
>
> [A2]. Thanks to the reviewer for the advice. The suggestion will make the article more convincing. We will make further improvements in this direction.
>
> [Q3]. Minor notes.
>
> [A3]. Thanks for the reviewer's comments. We will correct them and revise the sentences.

---

### Official Review · AnonReviewer4 · 2020-10-29
**Minimal Novelty but Good Results**

**Rating:** 5
**Confidence:** 4

**Review:**

This paper presents a novel technique for fine-grained visual classification. This technique addresses the classic issues in this task of inter-class similarity (coupled with intra-class variation) and the “long tailed” dataset problem, prevalent in datasets such as iNaturalist2018. The technique is an extension of Dubey et. al which extends their technique by incentivising the predictions for all samples in a mini-batch to be similar. This is in contrast to Dubey et. al which splits the mini-batch into two halves and incentivizes the aggregate predictions of the two halves to be similar.

I find this to be of minimal novelty, and I question the practicality of the method. Unless I have misunderstood, this sounds like it would be slower to train and I did not see any runtime analysis comparisons in the experiments section. Experiments showing how the new loss function effects training time would help alleviate this concern. The method does however improve performance on the tested benchmarks.

Finally, I found that there were numerous grammatical and stylistic mistakes. The writing improves during the discussion of the mathematics of the technique, but the introduction, experiments, and discussion need work. I would like to see the writing improved for a publication.

---

> ### Author Response · Authors · 2020-11-25
> **The novelty is that we propose a new confusion energy called BCN which can easily address the natural world data distribution problem with a competitive performance**
>
> [Q1]. I find this to be of minimal novelty, and I question the practicality of the method. Unless I have misunderstood, this sounds like it would be slower to train and I did not see any runtime analysis comparisons in the experiments section. Experiments showing how the new loss function effects training time would help alleviate this concern. The method does however improve performance on the tested benchmarks.
>
> [A1]. Firstly, we claim that the novelty of this paper is that our approach can solve the natural world data distribution with a general and reasonable form. Lots of previous works only focus on fine-grained or long-tailed problems. Second, we agree with the reviewer’s argument, but it won’t be the main issue. The runtime of our approach is only slower than the baseline but faster than all of the methods that are presented in Table 2. Hence, in a limited number of pages, we only claim that our training process is more simple and practical instead of presenting it as a table.
>
> [Q2]. Finally, I found that there were numerous grammatical and stylistic mistakes. The writing improves during the discussion of the mathematics of the technique, but the introduction, experiments, and discussion need work. I would like to see the writing improved for a publication.
>
> [A2]. We agree with the reviewer’s comments. In this paper, we straddle two different domains at the same time, and each of them is the mainstream research direction recently. Whether in fine-grained or long-tailed problems, each of them could have written as a complete article. Hence, in a limited number of pages, it is a little bit challenging to explain more details about how our method affects both of these problems. Finally, we will review and revise the writing as the reviewer mentioned.

---

### Official Review · AnonReviewer1 · 2020-10-29
**Interesting Idea but Not Convinced About the Method nor by the Experiments**

**Rating:** 4
**Confidence:** 5

**Review:**

**Overview:** The paper presents an extension to the Batch Confusion Norm (BCN) regularization technique so that it can account for imbalanced datasets. The extension to BCN implies adding a matrix that determines its values as a function of class imbalanced statistics contained in a batch. The paper presents experiments showing the benefits of the proposed extension on Fine-Grained Visual Classification (FGVC) and long-tailed (LT) tasks. The experiments show modest improvements over the baselines.

**Pros:**
*Clarity of the paper is good.* The clarity of the paper is very good. The motivation behind the FGVC and LT learning problems as well as the proposed method is clear. Thanks to the clarity of the paper I believe the reproducibility should be good. Also, I believe that the paper addresses an important problem with practical value.

**Cons:**
*Novelty.* The novelty of the paper IMHO falls short. This is because the submission mainly extends the BCN paper by adding a matrix A whose entries are a function of the statistics of the statistics of the dataset.  The proposed extension lacks a more rigorous derivation and justification; IMHO, the proposed extension seems to be ad hoc.

*Not convinced about BCN and proposed extension.*
1) I am not sure if BCN is a proper regularizer. While I understand the geometry behind minimizing the rank of the matrix P, I don't think this is a proper way of processing the columns of matrix P which are *posterior distributions*. I think the paper lacks a clear justification about using BCN as a way to treat posterior distributions without using statistical or probabilistic methods.
2) BCN is conditioned to operate if the classes in the batch are unique. However, satisfying this condition in practice can be challenging. This becomes challenging when dealing with long-tailed datasets. What is the optimal/efficient way to guarantee a batch that satisfies this constraint? If the constraint is not satisfied, can BCN provide bad gradients because repeated classes with different posteriors in different columns can double its contribution?
3) The extension of adding a matrix A is simple and feels a bit ad hoc. What is the intuition behind the matrix A in terms of the norm? Can the matrix A be considered a way to ` *weight* a posterior as a function of the statistics?

*Insufficient experiments and baselines.*
1) While I am pleased to see that the paper uses datasets depicting real scenes (e.g., iNaturalist, CUB, CAR, etc.), the experiments only focus on using one family of networks: ResNet. Does this method operate well on other more modern architectures, e.g., EfficientNet or MobileNetV2?
2) In terms of baselines, I think the paper is missing recent long-tail methods:
  a) Liu, et al. Large-Scale Long-Tailed Recognition in an Open World. CVPR 2019.
  b) Cao, et al. Learning Imbalanced Datasets with Label-Distribution-Aware Margin Loss. NeurIPS 2019.
3) The results for the long-tail methods are lacking more details. As it is common in various long-tail recognition papers, the paper is not showing performance on head or tail classes. It only shows an overall classification performance. This is misleading as the method can helping more head classes and thus improving the overall classification performance.

*Minor concern*: Most of the figures and diagrams look nearly identical to those presented in the BCM paper.

---

> ### Author Response · Authors · 2020-11-25
> **The novelty is we propose a confusion energy called ''BCN'' that can adaptively address the fine-grained and long-tailed distribution.**
>
> [Q1]. I am not sure if BCN is a proper regularizer. While I understand the geometry behind minimizing the rank of the matrix P, I don't think this is a proper way of processing the columns of matrix P which are posterior distributions. I think the paper lacks a clear justification about using BCN as a way to treat posterior distributions without using statistical or probabilistic methods.
>
> [A1]. Inter-class similarity between each category is the main challenging term in FGVC. Therefore, the proposed BCN is a confusion regularization that tries to equip the model with the ability to capture the inter-class similarity automatically. Hence, it is not a posterior distribution.
>
> [Q2]. BCN is conditioned to operate if the classes in the batch are unique. However, satisfying this condition in practice can be challenging. This becomes challenging when dealing with long-tailed datasets. What is the optimal/efficient way to guarantee a batch that satisfies this constraint? If the constraint is not satisfied, can BCN provide bad gradients because repeated classes with different posteriors in different columns can double its contribution?
>
> [A2]. The classes in a batch are unique is an assumption in the most extreme cases. On the other hand, few of classes in a batch have the same label is a normal condition. For instance, if two predictions have a similar probability distribution, the BCN loss between them becomes very small. In other words, same labels with small loss is intuitive. Hence, we do not make any restriction for generating each batch at the training time.
>
> [Q3]. The extension of adding a matrix A is simple and feels a bit ad hoc. What is the intuition behind the matrix A in terms of the norm? Can the matrix A be considered a way to ` weight a posterior as a function of the statistics?
>
> [A3]. The matrix A tries to control the confusion magnitude. For example, frequent categories has more instances to prevent a stronger confusion energy, and rare categories is opposite. Intuitively, we formulate the adaptive matrix A by the training data statistics. Hence, the matrix A is an adaptive confusion term, not a posterior mechanism.
>
> [Q4]. While I am pleased to see that the paper uses datasets depicting real scenes (e.g., iNaturalist, CUB, CAR, etc.), the experiments only focus on using one family of networks: ResNet. Does this method operate well on other more modern architectures, e.g., EfficientNet or MobileNetV2?
>
> [A4]. Yes, it still operates well on not only the ResNet series but also the EfficientNet, DenseNet, VGG, etc. In the powerful model, the results of EfficientNet-B7 from baseline, PC, and BCN approach are 70.9, 69.8, and 74.1, respectively.  The improvement of our approach is consistent with other networks. For the sake of the fair comparison to the other approaches, our main paper follows the most commonly used network, ResNet series, to report the experimental results.
>
> [Q5]. In terms of baselines, I think the paper is missing recent long-tail methods: a) Liu, et al. Large-Scale Long-Tailed Recognition in an Open World. CVPR 2019. b) Cao, et al. Learning Imbalanced Datasets with Label-Distribution-Aware Margin Loss. NeurIPS 2019.
>
> [A5]. We straddle two different domains at the same time, and each of them is the mainstream research direction recently. Hence, due to the limited space, we can only choose a few representative articles in their respective fields. a) has missed the experiments about iNaturalist which contains fine-grained and long-tailed properties both. We have the comparison with b) at Table 2 (b) LDAM and the citation is located in the second.
>
> [Q6]. The results for the long-tail methods are lacking more details. As it is common in various long-tail recognition papers, the paper is not showing performance on head or tail classes. It only shows an overall classification performance. This is misleading as the method can helping more head classes and thus improving the overall classification performance.
>
> [A6]. In Figure 3 (b), we have shown the performance on head or tail classes. More details can be found in the captions.
>
> [Q7]. Minor concern.
>
> [A7]. We can only say that this article is entirely our own work, without any plagiarism.

---

### Official Review · AnonReviewer2 · 2020-10-29
**Official Blind Review #2**

**Rating:** 5
**Confidence:** 5

**Review:**

This paper proposes the batch confusion norm (BCN) for dealing with both fine-grained recognition and long-tailed recognition simultaneously. Specifically, BCN considers the confusion regularization within each training batch and an adaptive matrix term is designed for handling the long-tailed problem. Experiments are conducted on fine-grained benchmark datasets (e.g., CUB, CAR, AIR) as well as long-tailed recognition datasets (e.g., iNat18).

Paper strengths:
- The paper is well organized and easy to follow.
- The problems studied in this paper, i.e., fine-grained recognition and long-tailed visual recognition, are both important, challenging and practical in computer vision, which deserves further studies.
- The proposed method sounds reasonable.

Paper weaknesses:
- Although the proposed method is reasonable, some specific model designs are not quite clear. 1) Regarding Eq. (2), the reason why it requires to optimize the ranking should be further explained and its motivation needs to state. 2) Regarding Eq. (5), what the intuition of the adaptive matrix (i.e., (log_{\mu+1} (N_i+1))^{\delta^{\tau}}) when i = j should be provided to the authors.
- The major issue of this paper is the experimental evaluations. 1) The classification accuracy on these fine-grained benchmark datasets and iNat18 are not significantly better than the accuracy of previous work. Thus, the effectiveness of the proposed method is problematic. 2) Some state-of-the-art methods are not involved in the experimental comparisons, such as [ref1-ref5]. Moreover, the accuracy of the proposed method cannot outperform these methods.

Minor issues:
- There are several typos and writing problems in this paper. For example, on Page 3, "Dubey et al.Dubey et al. (2018)", and "Chen et al.Chen et al. (2019)". On Page 4, "PC Dubey et al. (2018)". On Page 8, "And also solves the long-tailed problem by an adaptive matrix term."

[ref1] Weakly Supervised Fine-grained Image Classification via Guassian Mixture Model Oriented Discriminative Learning, CVPR 2020.

[ref2] Weakly Supervised Complementary Parts Models for Fine-Grained Image Classification from the Bottom Up, CVPR 2019.

[ref3] Fine-Grained Visual Classification via Progressive Multi-Granularity Training of Jigsaw Patches, ECCV 2020.

[ref4] Learning Imbalanced Datasets with Label-Distribution-Aware Margin Loss, NeurIPS 2019.

[ref5] BBN: Bilateral-Branch Network with Cumulative Learning for Long-Tailed Visual Recognition, CVPR 2020.

---

> ### Author Response · Authors · 2020-11-25
> **Our general and simple approach solves the fine-grained and long-tailed distribution at the same time with competitive results.**
>
> [Q1]. Although the proposed method is reasonable, some specific model designs are not quite clear. 1) Regarding Eq. (2), the reason why it requires to optimize the ranking should be further explained and its motivation needs to state. 2) Regarding Eq. (5), what the intuition of the adaptive matrix (i.e., ($log_{\mu+1} (N_i+1))^{\delta^{\tau}}$) when i = j should be provided to the authors.
>
> [A1]. 1) In our paper, we try to address the fine-grained and long-tailed problems at the same time. The point of view we choose is “confusion energy”. The main idea of confusion energy is making the model learn the inter-class similarity and prevent the overfitting issue. Moreover, different from pairwise confusion (PC), we make the confusion efficiently from the prediction of each batch. Hence, our method starts by minimizing the rank of the prediction matrix. 2) The adaptive matrix A tries to control the confusion magnitude for each category. Hence, when $i = j$, it means that we only control the confusion magnitude on each class individually.
>
> [Q2]. The major issue of this paper is the experimental evaluations. 1) The classification accuracy on these fine-grained benchmark datasets and iNat18 are not significantly better than the accuracy of previous work. Thus, the effectiveness of the proposed method is problematic. 2) Some state-of-the-art methods are not involved in the experimental comparisons, such as [ref1-ref5]. Moreover, the accuracy of the proposed method cannot outperform these methods.
>
> [A2].  1) Firstly, we straddle two different domains at the same time, and each of them is the mainstream research direction recently. Second, we achieve a competitive performance without any data sampling methods or any additional model parameters. Third, we solve the problems by a very special point, confusion energy. Our significant improvement is against PC or baseline. In conclusion, this paper tries to introduce to everyone that we can address fine-grained and long-tailed problems by the confusion energy way. Furthermore, the training process shows that it is practical and the experimental results present the competitive performance. 2) We solve the problems from a very different concept. [ref1-ref3] try to capture the discriminative parts by the network design. All of them need a larger model size and a complex training process. [ref4-ref5] only focus on the long-tailed problem. And the performance in ref4, as we all know, is hard to reproduce.
>
> [Q3]. Minor issues.
>
> [A3]. Thanks for the mistakes picked out by the reviewer, we will correct them.

---

### Decision · Program_Chairs · 2021-01-07
**Final Decision**

**Decision:**

Reject

**Comment:**

The authors address the problem of fine-grained image classification. They propose a batch based regularizer, called the batch confusion norm (BCN), to encourage less over confident predictions. They also tackle the problem of class imbalance during training by adaptively weighting the BCN loss at the class level to take the imbalances in the underlying label distributions into account. Results are presented on four different fine-grained datasets.

Overall, while the reviewers had some positive comments, there was not broad support for the paper. There are questions that need to be resolved related to the evaluation e.g. the best performing model uses GASPP, however there is no reported GASPP variant for the PC baseline. Similarly, it would be valuable to know how much PC would benefit from an additional class imbalance term in the iNaturalist2018 results. Given that the proposed regularizer builds on PC (Dubey et al.), it is very important that the authors provide a like-for-like comparison so that readers can better understand the merits of the proposed method.

There were also concerns with the presentation of the paper e.g. several typos (which can be easily fixed), issues with the clarity of the text (which require more work), and uninformative figures (e.g. Fig 2 should be revised to more clearly illustrate the differences between the three methods shown). The authors are encouraged to revise the text to resolve these problems.

While the paper has some strengths (e.g. the empirical performance on some of the tasks is promising and the method is conceptually simple), there are still a number of concerns from the reviewers e.g. a lack of a clear motivation as to why the proposed method works, and why it is conceptually better than existing alternatives (e.g. PC). Given this lack of support, it is not possible to recommend the paper in its current form.